# *PTCH1* Gene Variants, mRNA Expression, and Bioinformatics Insights in Mexican Cutaneous Squamous Cell Carcinoma Patients

**DOI:** 10.3390/biology13030191

**Published:** 2024-03-16

**Authors:** Marianela Zambrano-Román, Jorge R. Padilla-Gutiérrez, Yeminia Valle, José Francisco Muñoz-Valle, Elizabeth Guevara-Gutiérrez, Patricia Aidé López-Olmos, Laura Cristina Sepúlveda-Loza, Luis Alberto Bautista-Herrera, Emmanuel Valdés-Alvarado

**Affiliations:** 1Instituto de Investigación en Ciencias Biomédicas (IICB), Centro Universitario de Ciencias de la Salud, Universidad de Guadalajara, Guadalajara 44340, Mexico; marianela.zambrano3682@alumnos.udg.mx (M.Z.-R.); ramon.padilla@academicos.udg.mx (J.R.P.-G.); yeminia.valle@academicos.udg.mx (Y.V.); drjosefranciscomv@cucs.udg.mx (J.F.M.-V.); 2Doctorado en Genética Humana, Departamento de Biología Molecular y Genómica, Universidad de Guadalajara, Guadalajara 44340, Mexico; 3Departamento de Dermatología, Instituto Dermatológico de Jalisco “Dr. José Barba Rubio”, Secretaría de Salud Jalisco, Zapopan 45190, Mexico; albtlacuilo@yahoo.com (E.G.-G.); paty.olmos39@gmail.com (P.A.L.-O.); 4Departamento de Dermatología, Hospital Civil de Guadalajara “Fray Antonio Alcalde”, Guadalajara 44200, Mexico; lau.derma.92@gmail.com; 5Departamento de Farmacobiología, Centro Universitario de Ciencias Exactas e Ingenierías, Guadalajara 44430, Mexico; luis.bautista4106@academicos.udg.mx

**Keywords:** cutaneous squamous cell carcinoma, skin cancer, non-melanoma skin cancer, *PTCH1*, genetic variants, mRNA, bioinformatics

## Abstract

**Simple Summary:**

Skin cancer represents a prevalent and significant global health concern. Our study focused on comprehending cutaneous squamous cell carcinoma in the Mexican population through the examination of genetic variants of the *PTCH1* gene, the assessment of its mRNA’s relative expression, and the utilization of bioinformatics tools. Despite the absence of statistically significant associations in the experimental results, integrating bioinformatics was imperative to understand the impact of the analyzed variants on protein function, splicing, expression, and disease-related aspects. This contributes to our overarching comprehension of non-melanoma skin cancer development. Further investigation is crucial to gain a comprehensive understanding of the in vivo role of these genetic variants.

**Abstract:**

Background: Skin cancer is one of the most frequent types of cancer, and cutaneous squamous cell carcinoma (cSCC) constitutes 20% of non-melanoma skin cancer (NMSC) cases. *PTCH1*, a tumor suppressor gene involved in the Sonic hedgehog signaling pathway, plays a crucial role in neoplastic processes. Methods: An analytical cross-sectional study, encompassing 211 cSCC patients and 290 individuals in a control group (CG), was performed. A subgroup of samples was considered for the relative expression analysis, and the results were obtained using quantitative real-time PCR (qPCR) with TaqMan^®^ probes. The functional, splicing, and disease-causing effects of the proposed variants were explored via bioinformatics. Results: cSCC was predominant in men, especially in sun-exposed areas such as the head and neck. No statistically significant differences were found regarding the rs357564, rs2236405, rs2297086, and rs41313327 variants of *PTCH1*, or in the risk of cSCC, nor in the mRNA expression between the cSCC group and CG. A functional effect of rs357564 and a disease-causing relation to rs41313327 was identified. Conclusion: The proposed variants were not associated with cSCC risk in this Mexican population, but we recognize the need for analyzing larger population groups to elucidate the disease-causing role of rare variants.

## 1. Introduction

Non-melanoma skin cancer (NMSC) stands as one of the most prevalent forms of cancer globally, with cutaneous squamous cell carcinoma (cSCC) accounting for approximately 20% of all NMSC cases, placing it as the second most common type of epithelial malignancy. The incidence of skin cancers continues to increase, and it is well-known that chronic exposure to UV radiation (UVR) from sunlight is the main risk factor [1,2]. cSCC arises from epidermal keratinocytes, and non-UVR risk factors such as older age, fair skin, immunosuppression, and exogenous chemical mutagens are relevant in its development [3,4,5]. Histologically classified according to differentiation degree, cSCC carries a risk of metastasis around 2–4%, with specific factors such as the site, diameter, and differentiation influencing local recurrence and metastatic potential [6,7].

Sonic hedgehog is an evolutionarily conserved pathway highly activated during embryonic development and predominantly inactivated in adults. This signaling pathway regulates proliferation, which is essential for maintaining stem cells for tissue repair, matrix remodeling, and angiogenesis in tissues such as the skin and epithelial tissues of internal organs. However, reactivation of the signaling may play a significant role in carcinogenesis processes [8,9,10]. The *PTCH1* gene is a tumor suppressor gene that encodes for the PTCH1 protein, a key element of this pathway acting as a ligand receptor that downregulates the signaling. Genetic variants, mRNA expression dysregulation, and epigenetic modifications in *PTCH1* have been described across diverse cancer types [11,12,13,14,15].

Research on skin cancer has predominantly focused on basal cell carcinoma (BCC), which stands as the most prevalent non-melanoma skin cancer (NMSC). This emphasis is attributed to the association between *PTCH1* and Gorlin syndrome. It is estimated that 60–70% of individuals with classical manifestations of Gorlin syndrome harbor germline pathogenic variants in the *PTCH1* gene, with most of these variations leading to a loss of protein function. These patients typically present with multiple BCCs, resulting in higher recurrence rates [16,17]. While the role of *PTCH1* in cutaneous squamous cell carcinoma (cSCC) has not been fully elucidated, it is well established that nearly all BCCs exhibit activated hedgehog signaling due to genetic variants in the *PTCH1* gene [18]. Additionally, differential expression analysis has revealed high *PTCH1* expression in BCC but suppressed expression in cSCC [18].

The *PTCH1* variants rs357564 and rs2236405 have been analyzed in other types of cancer and congenital diseases, and their relevance may be associated with their location in the protein structure and its interaction with the Smoothened protein. In contrast, the rs2297086 variant has been studied in patients with non-melanoma skin cancer post transplant, while the importance of the rs41313327 variant may stem from its location within one of the extracellular loops critical for receptor–ligand interaction [19,20,21,22,23]. This study aimed to analyze the distribution of the allele and genotype frequencies among Mexican patients with cSCC, as well as the expression of their mRNA, and the possible biological, splicing, and expression effects through bioinformatics tools.

## 2. Materials and Methods

### 2.1. Subjects

The study group comprised 211 patients from western Mexico with histopathological diagnoses of cutaneous squamous cell carcinoma (cSCC) recruited from the Instituto Dermatológico de Jalisco “Dr. José Barba Rubio”. The control group (CG) consisted of 290 individuals. We considered only unrelated individuals with three previous generations, including their own, who were born in this region of Mexico. The relative expression of their mRNA was analyzed in 26 samples from the cSCC patients and 44 samples from the CG.

### 2.2. Genotyping of PTCH1 Variants

Genomic DNA was extracted from peripheral blood leukocytes using Miller’s technique [24]. The analysis of the variants was performed with quantitative real-time PCR (qPCR) utilizing TaqMan probe assays: rs41313327 (C__86344157_10), rs357564 (C___3030099_10), rs2297086 (C__16185796_10), and rs2236405 (C__15954310_10), and TaqMan Genotyping Master Mix catalog 4371355 (Applied BiosystemsTM, Foster City, CA, USA) with a Roche LightCycler 96^®^ System. The sample size was calculated using OpenEpi software version 3.017 (https://www.openepi.com/Menu/OE_Menu.htm, accessed on 15 January 2023), considering a confidence level of 95% (1—α) and a statistical power of 80%. A minor allele frequency (MAF) of 0.20 for the rs2297086 variant was employed as the hypothetical proportion of controls with exposure. This frequency was reported in Latin American individuals with predominantly European and Native American ancestry from The ALFA Project, NCBI. A double-blind genotyping of 25% of the analyzed samples was performed for all variants as quality control. The agreement rate was 100%, validating the genotyping.

### 2.3. Relative Expression of the mRNA Analysis

Total RNA was extracted from peripheral blood leukocytes using TRI Reagent^®^ (SIGMA-ALDRICH, St. Louis, MO, USA) to obtain total RNA based on the method of Chomczynski and Sacch [25]. Subsequently, cDNA was synthesized through reverse transcription, utilizing 1 µg of total RNA in accordance with the manufacturer’s instructions. The reverse transcription reagents included M-MLV Reverse Transcriptase, dNTP Mix, oligo (dT) 15 Primer, RNasin^®^ Ribonuclease Inhibitor, and Ribonuclease H (Promega Corporation, Madison, WI, USA). Relative expression was assessed with qPCR utilizing TaqMan probes *PTCH1* (catalog 4448489) and *GAPDH* (catalog 4331182), with the latter serving as a normalizer. The results were analyzed using the 2^−ΔΔCq^ method for relative fold change and the 2^−ΔCq^ method as the unit relative of expression (URE), as reported by Schmittgen and Livak [26]. For the analysis of relative expression, a convenience sample size was considered, with careful contemplation of age, sex, and genotypes to ensure representativeness across all study groups and their analyzed characteristics.

### 2.4. Bioinformatics

The bioinformatics analysis encompassed exonic (rs357564, rs2236405, rs41313327) and intronic (rs2297086) variants. Multiple perspectives were considered, including homology-based predictions utilizing the Sorting Intolerant from Tolerant (SIFT), Protein Variation Effect Analyzer (PROVEAN), and Functional Analysis through Hidden Markov Models (FATHMM) software v2.3. These tools assess amino acid substitutions and their impact on protein function [27,28,29]. Additionally, structural homology-based prediction was performed using the Polymorphism Phenotyping-2 (Polyphen-2) software v2 [30]. To predict the conserved score of amino acid residues and the altered molecular mechanisms for amino acid substitutions, the ConSurf and MutPred2 web servers were employed [31,32]. Furthermore, the prediction of disease-related variants and the pathogenicity of rare missense variants were determined using SNP&GO and Rare Exome Variant Ensemble Learner (REVEL), respectively [33,34]. For the intronic variant, the Human Splicing Finder (HSF) was utilized to predict splicing signals or identify splicing motifs [35], and the regSNP-intron was employed to detect the disease-causing potential of single nucleotide variants (SNVs) [36].

### 2.5. Statistical Analyses

The binary logistic regression analysis was conducted to ascertain the risk associated with demographic characteristics in the development of cSCC, as well as the clinical features of patients in the development of lesions with greater histopathological invasion. The Hardy–Weinberg equilibrium test, genotypes, and allele frequencies were calculated using the χ^2^ or Fisher’s exact test, as applicable. Lewontin’s D’ was utilized to evaluate the linkage disequilibrium (LD) [37]. The haplotype analysis was performed using SNPStats web tool (https://www.snpstats.net/start.htm?, accessed on 8 May 2023) and SHEsis web-based platform (http://analysis.bio-x.cn/myAnalysis.php, accessed on 8 May 2023) [38,39]. Odds ratios (ORs) and 95% confidence intervals (CIs) were calculated to test the probability that the genotype and allele frequencies were associated with cSCC. A *p*-value < 0.05 was considered statistically significant. All of the statistical analyses were conducted with the SPSS statistical package version 20.

## 3. Results

### 3.1. Population Description

The demographic characteristics of the studied groups are presented in Table 1. In this cohort, male patients comprised 59.2%, while females accounted for 40.8% of the cases. The median ages were 72 (interquartile range 65–80 years) and 66 (interquartile range 58–72 years) for cSCC patients and the control group (CG), respectively.

The binary logistic regression analysis revealed that being male (OR = 1.81, 95% CI 1.26–2.62, *p* = 0.002) and aged over 70 years (OR = 2.31, 95% CI 1.60–3.34, *p* < 0.001) are significant risk factors for the development of cSCC, as evidenced in the comparison between the control group (CG) and the cSCC patients (Table 2).

Regarding the clinical and histological characteristics of the cSCC patients, the primary location was in the head and neck of men and women, with a size of less than 2 cm. Most of the lesions were classified as invasive; however, the majority had a high differentiation degree. Statistically significant differences were observed regarding the location between men and women of the cSCC group (*p* < 0.05) (Table 3).

In order to analyze the association of clinical characteristics in the development of invasive lesions, the logistic regression model confirmed that males continue to have the higher risk (OR = 2.09, 95% CI 1.04–4.22, *p* = 0.038), along with the higher occurrence of lesions in sun-exposed areas such as the head, neck, and arms (OR = 2.09, 95% CI 4.12–23.35, *p* = < 0.001) (Table 4).

### 3.2. PTCH1 Genetic Variant Genotyping

PTCH1 variants rs357564, rs2236405, rs2297086, and rs41313327 were analyzed in the cSCC patients and the CG. The variants rs357564 and rs2236405 were found to conform to Hardy–Weinberg equilibrium (HWE), while rs2297086 and rs41313327 did not (*p* < 0.05), according to previously published results for our control group [40]. This is related to the fact that only the major alleles of the rs2297086 (G) and rs41313327 (C) variants were present in all of the genotyped individuals. Both dominant (GG vs. GA + AA) and recessive (GG + GA vs. AA) models were derived only from rs357564. The A/A genotype of the rs2236405 variant was not present in the studied groups. However, the frequency of the A allele aligned with previous reports. There was no inheritance model, genotype, or allele association between PTCH1 variants and the risk of developing cSCC. The allele and genotype frequencies of the rs357564 and rs2236405 variants are shown in Table 5.

### 3.3. Haplotype Analysis of PTCH1

Only the variants rs357564 and rs2236405 were considered for this analysis and were found in moderate linkage disequilibrium (LD) (D’ = 0.77, *p* = 0.0047). The prevailing haplotype was GT, accounting for 50.3% in the cSCC patients and 49.8% in the control group (CG), respectively. However, no statistically significant association was identified between the haplotypes and the risk of cSCC (Table 6).

### 3.4. Relative Expression of the PTCH1 mRNA

The expression of the PTCH1 mRNA was identified in 6 of the 26 analyzed cSCC samples, and 14 of the 44 samples from the CG. There were no statistically significant differences regarding the presence or absence of expression of the gene of interest when compared by clinical features such as age, sex, location, and the size of the lesions, and histopathological classification. The cSCC patients presented a lower median mRNA expression from their blood leukocytes compared to the CG (2.1-fold less). Nonetheless, there were no statistically significant differences while evaluating the data through the 2^−ΔCq^ method (0.96 URE vs. 2.81 URE, respectively; *p* = 0.21) (Figure 1).

### 3.5. Bioinformatics Analysis

Several bioinformatics tools were employed to investigate the effects of the SNVs rs357564, rs2236405, and rs2297086 on the protein function, as well as the effect of rs2297086 on splicing. Regarding the functional effect, we could attribute a potential pathogenic effect to the rs357564 variant. However, for the rs223405 and rs41313327 variants, the results were inconclusive, classifying them as either benign or possibly pathogenic. Notably, for rs41313327, a potential effect on a molecular mechanism affecting a neighboring residue was identified. In terms of pathogenicity, this variant is reported as apparently disease-causing, being the only one within the analyzed group. The intronic variant rs2297086 was classified as benign, as no impact on splice signals, such as donor and/or acceptor sites, branching points, or enhancer/silencer elements, was detected. It is essential to highlight that the variability in predictions arises from the different types of alignments and features evaluated with each platform (Table 7).

According to the results obtained from Consurf, and the analysis of conserved residues in the PTCH1 protein, we observed that the proline at position 1315 was considered a variable residue, and the change may not have had a functional effect on the protein. Nonetheless, this residue was located immediately after another proline classified as conserved, and both residues were indicated to be exposed. On the other hand, the threonine residue at position 1195 of the protein was not classified as variable or conserved but was deemed as average, which may not be conclusive. To conclude, the aspartic acid at position 850 was described as a moderately conserved residue, which is in contrast with the results of the previously utilized platform, possibly indicating a likely functional effect (Figure 2). The result of the complete protein sequence analysis can be found in Appendix A.

## 4. Discussion

### 4.1. Clinical and Histopathological Features

The incidence of NMSC continue to rise, with prolonged exposure to UV radiation identified as the primary risk factor. cSCC is the second most common type of NMSC, and has been reported to exhibit higher aggressiveness and metastatic potential [41,42]. The analyzed cSCC group of patients was mainly represented by men, and the age of onset continues to be primarily between the sixth and seventh decades of life, according to previous reports [43,44]. However, it is necessary to consider that sex comparisons should be approached carefully due to external factors such as occupation, socioeconomic level, and sunscreen use among different populations worldwide [45,46].

For several decades, the predominant observation has been that cSCC typically manifest in sun-exposed areas such as the head, neck, and arms—a pattern consistent with our patient cohort [42,47,48]. This is relevant since most cases of metastasis are reported in these regions, in proximity to regional lymph nodes [49]. Notably, lesions exceeding 2 cm in size are associated with a twofold increase in recurrence risk and a threefold increase in metastatic potential [7]. The current study group showed lesions with a size less than 2 cm, and they were mainly classified as being well differentiated, considering the nuclear atypia and keratinization degree. This characterization aligns with a low-risk profile and serves as a favorable prognostic factor, as outlined by Parekh and Seykora [50].

### 4.2. PTCH1 Variants and mRNA Expression

Cell proliferation involves intricate molecular mechanisms, including the Sonic hedgehog signaling pathway. The PTCH1 protein is responsible for negatively controlling pathway activation, primarily in embryogenesis, and progressively decays in adulthood. However, embryogenesis and carcinogenesis share common features related to cellular processes such as proliferation, differentiation, and migration. Disruption of function due to genotypic alterations may contribute to the development of more aggressive cancer types [51]. The identification of aberrant hedgehog signaling has, in fact, paved the way for therapeutic advancements in other forms of NMSC [12]. The role of PTCH1 in cSCC has been poorly described compared to BCC [52].

Even though phenotypic characteristics have been broadly described, the genetic structure of the population has rarely been highlighted, particularly in the quest to establish correlations between prevalence and geographic distribution [53]. Analyses of the rs357564 variant have been performed in several types of cancer and clinical conditions [54,55,56,57]. Most of the reports on rs357564, as well as the rest of the variants proposed in this investigation, have been related to BCC. Nonetheless, Asplund et al. [19] described a distinct prevalence of the G/A genotype in cSCC patients compared to the G/G genotype observed in BCC patients.

Although intronic variants are densely distributed in the genome, only a limited portion has been evaluated for alterations of biological functions [35]. There are few reports about the rs2297086 variant, and the impact of such variants on gene expression may occasionally appear ambiguous. They potentially perform a regulatory role, especially when their positioning is distant from the exon–intron boundary, which could lead to the generation of alternative splicing sequences [58]. Nevertheless, for this variant, a benign effect is predicted because it is located in a poorly conserved position (ClinVar Miner, dbSNP: rs2297086) [59].

The frequency of the homozygous genotype for the wild-type allele (C/C) in the rs41313327 (C>T) variant was identified in both the cSCC group and the CG, in accordance with prior database reports. One previous analysis of patients with non-cancerous conditions in a Turkish population did not reveal the existence of the minor allele (T) [23]. In another previous report in a Polish population of patients with BCC, the C/C, C/T, and T/T genotypes were found, although without statistically significant differences between the patients and controls [60]. Similarly, the minor allele (A) of the rs2236405 variant has been documented with a low frequency, consistent with our results, in which the A allele was only identified in a few heterozygotes (T/A). Despite this, the rs2236405 variant is classified as benign considering its population frequency, intact protein function, and lack of association with disease in case–control studies, so it has not been associated with a known cause of pathogenicity (ClinVar Miner, dbSNP: rs2236405).

The rs357564 and rs2236405 variants were found to be in moderate LD (D’ = 0.77, *p* = 0.0047), but there was no association between the haplotypes and the cSCC risk, consistent with our previous report in BCC [40]. As non-canonical mechanisms are known to activate HH signaling, it is plausible that other mechanisms are involved in the role of PTCH1 in tumorigenesis, not specifically related to these variants [12]. Regarding the relative expression analysis, we consider that peripheral leucocytes may not be appropriate since the expression levels seem to be too low in this matrix, especially in the cSCC patients, where a low number of samples showed a detectable expression. Additionally, we recognize a trend towards lower expression in patients compared to controls, aligning with the known function of *PTCH1* and its association with cancer.

### 4.3. Bioinformatics Analysis

The progress in sequencing projects has facilitated a massive generation of data, providing an opportunity to assess variations in genomes. Bioinformatics tools allow predictions to be made about functional effects or splicing mechanisms through alignments between homologous sequences, assuming that the evolutionary conservation of specific amino acids defines them as functionally important. Nonetheless, these tools frequently differ due to their operation with different algorithms and predictive models [27,32,34,61,62].

In order to comprehend the role of these variants, we recognize the increasing significance of utilizing in silico tools. This is driven by the potential effects arising from amino acid changes in the protein sequence. However, the absence of clinical information may pose a limitation in determining the disease association of these variants. Consequently, there is a suspicion that conserved protein residues might contribute to altered or even loss of protein function [31]. Understanding the biological distinctions between rare (low-frequency) and common variants is crucial, as it can impede the differentiation of neutral rare variants from potentially pathogenic ones. A precise discrimination of rare neutral variants would necessitate an allele frequency higher than 3% [34].

As the rs357564 variant is situated in the C-terminal domain, and based on our findings, we propose that it has a potential effect on protein function. Hydrophobic residues, as shown in previous studies, play a crucial role in stabilizing and folding proteins [63]. However, we did not identify specific disease-related pathogenicity for this variant. Conversely, precise functional effects for the rs41313327 and rs2236405 variants have not yet been defined.

MutPred2 did not provide results for the molecular mechanism of rs357564 and rs2236405. However, it offered insights into rs41313327, despite it having the lowest frequency among the analyzed variants. According to these findings, the amino acid change of aspartic acid to asparagine at residue 850 may lead to an altered transmembrane protein or impact the sulfation—a common post-translational modification—of a neighboring tyrosine. From this perspective, this change might not directly affect the protein function, but it could potentially alter other residues. Significantly, these residues are located in the extracellular loops, which are associated with protein–ligand interactions. Moreover, the rs41313327 variant was the only variant reported as likely being disease-causing in the REVEL results, which demonstrated superior performance in distinguishing pathogenic variants from rare neutral variants with low allele frequencies [34]. This underscores the importance of analyzing and elucidating the role of rare variants as a potential cause of complex diseases [64].

Previous reports indicated a possible effect of missense variants in the *PTCH1* gene, but did not provide conclusive support regarding the harmful impact [65]. It is crucial to emphasize that bioinformatics analysis provides a diverse range of perspectives, and the interpretation of such results must be approached with caution. In vivo analysis and experimental evidence should always be considered as primary sources of knowledge.

We acknowledge that the sample size used in this study presented a limitation for detecting the minor allele in the rare variants analyzed (rs2297086 and rs41313327). However, we recognize that the presence of rare variants in the genome could contribute to susceptibility to common diseases [64]. Additionally, the selection of the appropriate tissue for conducting relative expression analysis is crucial in the accurate measurement of the mRNA of interest and its association with the studied disease. The analysis of mRNA from peripheral blood cells may be our primary limitation for studying *PTCH1* mRNA in this cohort, resulting in a lack of evidence to associate mRNA levels with the development of cSCC. Similarly, it is not feasible to ascertain whether the analyzed variants may exert an impact on transcriptional regulation and, consequently, on mRNA expression.

Despite these limitations, we perceive this work as a valuable starting point for future research. Analyzing larger sample sizes and conducting comparisons between tumor tissue and healthy skin, as well as engaging in functional studies, could contribute valuable insights to the study of the *PTCH1* gene and its role in cSCC.

## 5. Conclusions

In conclusion, although the proposed variants were not found to be associated with cutaneous squamous cell carcinoma, we cannot dismiss the necessity of analyzing larger population groups to elucidate the role of rare variants and potentially establish a connection between *PTCH1* variants and the development of cSCC, as it has been demonstrated with other types of cancer. Despite the mentioned limitations, we consider that bioinformatics platforms are valuable tools that allow us to formulate hypotheses, which we hope can be corroborated experimentally in the future.

## Figures and Tables

**Figure 1 biology-13-00191-f001:**
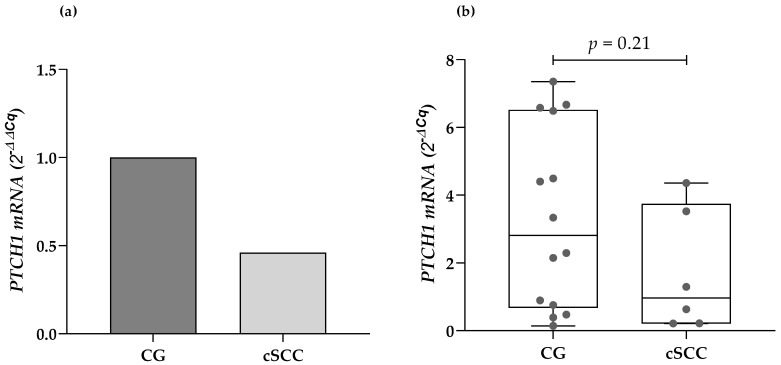
Comparison of the PTCH1 mRNA expression in cSCC patients and the CG using (**a**) the 2^−ΔΔCq^ method and (**b**) the 2^−ΔCq^ method. cSCC: cutaneous squamous cell carcinoma, CG: control group.

**Figure 2 biology-13-00191-f002:**
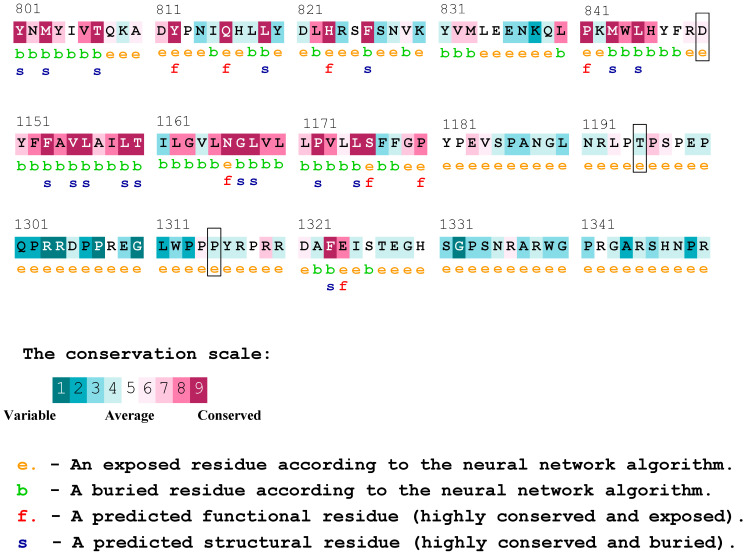
Consurf conserved residues for the PTCH1 protein. The residues affected by the variants rs357564 (P1315L), rs2236405 (T1195S), and rs41313327 (D850N) are highlighted in boxes.

**Table 1 biology-13-00191-t001:** Demographic characteristics of the cSCC patients and CG.

Variable	cSCC (*n* = 211) *n* (%)	CG (*n* = 290) *n* (%)	*p*
Age (years)	^a^ 72 (65–80)	^a^ 66 (58–72)	**<0.0001**
Sex			
Male	125 (59.2)	132 (45.5)	**0.002**
Female	86 (40.8)	158 (54.5)	

^a^: the data were expressed as the median and interquartile range (Q25–Q75). cSCC: cutaneous squamous cell carcinoma, CG: control group. Results with significant *p*-values are emphasized in bold.

**Table 2 biology-13-00191-t002:** Risk factors for developing cSCC.

Independent Variables	β ^a^	S.E. ^b^	Wald ^c^	D.F ^d^	*p*-Value	OR ^e^	95% CI ^f^	
							Low	High
Sex								
Female (ref)								
Male	0.594	0.188	10.043	1	**0.002**	1.81	1.26	2.62
Age								
<70 years (ref)								
≥70 years	0.838	0.187	20.017	1	**<0.001**	2.31	1.60	3.34

^a^: regression coefficient, ^b^: standard error, ^c^: Wald test, ^d^: degrees of freedom, ^e^: odds ratio, ^f^: confidence interval. Significant *p*-values are emphasized in bold.

**Table 3 biology-13-00191-t003:** Clinical and histological characteristics of cSCC patients.

Variable	M	F	*p*
	*n* (%)	*n* (%)	
Classification			
In situ	26 (20.8)	27 (31.4)	0.064
Invasive			
High	87 (69.6)	52 (60.5)	
Intermediate	12 (9.6)	5 (5.8)	
Low	-	2 (2.3)	
Size			0.961
<2 cm	76 (60.8)	52 (60.5)	
>2 cm	49 (39.2)	34 (39.5)	
Location			
Head and neck	84 (67.2)	52 (60.5)	**0.045**
Trunk	18 (14.4)	8 (9.3)	
Arms	15 (12)	19 (22.1)	
Legs	4 (3.2)	7 (8.1)	
More than 1 lesion	4 (3.2)	-	

M: male, F: female. Results with significant *p*-values are emphasized in bold.

**Table 4 biology-13-00191-t004:** Risk for developing invasive cSCC.

Independent Variables	β ^a^	S.E. ^b^	Wald ^c^	D.F ^d^	*p*-Value	OR ^e^	95% CI ^f^	
							Low	High
Sex								
Female (ref)								
Male	0.740	0.357	4.296	1	**0.038**	2.09	1.04	4.22
Age								
<60 years (ref)			4.302	2	0.116			
60–69 years	−0.258	0.625	0.170	1	0.680	0.773	0.22	2.63
≥70 years	−0.944	0.565	2.788	1	0.095	0.389	0.13	1.18
Location								
Not sun-exposed (ref)								
Sun-exposed	2.284	0.442	26.660	1	**<0.001**	9.814	4.12	23.35
Size								
<2 cm (ref)								
>2 cm	0.460	0.381	1.457	1	0.227	1.584	0.75	3.34

^a^: regression coefficient, ^b^: standard error, ^c^: Wald test, ^d^: degrees of freedom, ^e^: odds ratio, ^f^: confidence interval. Not-sun-exposed areas include the legs and trunk; sun-exposed areas include the head/neck and arms. Adjusted by histopathological classification: 0 = in situ and 1 = invasive. Significant *p*-values are emphasized in bold.

**Table 5 biology-13-00191-t005:** Allele and genotype distribution of the *PTCH1* gene variants in cSCC patients and the CG.

**Variant**	**CG**	**cSCC**	**OR**	
**rs357564**	***n* = 290 (%)**	***n* = 211 (%)**	**CI (95%)**	** *p* **
*Genotype*				
G/G °	73 (25.2)	59 (27.9)	−1	-
G/A	149 (51.4)	101 (47.9)	0.84 (0.55–1.28)	0.42
A/A	68 (23.4)	51 (24.2)	0.93 (0.56–1.53)	0.77
*Allele*				
G	295 (50.8)	219 (51.9)		
A	285 (49.2)	203 (48.1)	0.96 (0.74–1.23)	0.75
**Variant**	**CG**	**cSCC**	**OR**	
**rs2236405**	***n* = 290(%)**	***n* = 211(%)**	**CI (95%)**	** *p* **
*Genotype*				
T/T °	284 (97.9)	203 (96)	1	-
T/A	6 (2.1)	8 (4)	1.86 (0.64–5.46)	0.25
A/A	0 (0)	0 (0)	-	-
*Allele*				
T	574 (99)	414 (98)		
A	6 (1)	8 (2)	1.84 (0.64–5.36)	0.26

Pearson’s Chi square (χ^2^), odds ratio (OR), confidence interval (CI). cSCC: cutaneous squamous cell carcinoma, CG: control group. ° Reference category.

**Table 6 biology-13-00191-t006:** Distribution of the *PTCH1* haplotypes in the cSCC patients and the CG.

	cSCC	CG		
Haplotype	*n* (%)	*n* (%)	OR (CI 95%)	*p*
GT	212 (50.3)	289 (49.8)	1	-
AT	205 (48.6)	285 (49.1)	1.04 (0.81–1.35)	0.75
GA	5 (1.3)	6 (1.1)	0.61 (0.20–1.90)	0.40

cSCC: cutaneous squamous cell carcinoma; CG: control group; CI: confidence interval; OR: odds ratio. Haplotypes are represented by the rs357564 and rs2236405 variants.

**Table 7 biology-13-00191-t007:** Bioinformatics analysis of *PTCH1* variants.

SNV	Classification	Amino Acid Change	SIFT	PROVEAN	Polyphen-2	FATHMM	MutPred2	SNP&GO	REVEL
			Predictions						
**rs357564**	Missense variant	P1315L	Deleterious Score 0.023	Benign Score −1.39	Possibly damaging Score 0.711	Deleterious Score −2.76	-	Neutral	Likely benign Score 0.443
**rs2236405**	Missense variant	T1195S	Tolerated Score 0.238	Benign Score −1.78	Possibly damaging Score 0.469	Deleterious Score −2.71	-	Neutral	Likely benign Score 0.432
**rs41313327**	Missense variant	D850N	Tolerated Score 0.134	Deleterious Score −3.07	Benign Score 0.141	Deleterious Score −2.47	* Altered transmembrane protein ** Loss of sulfation at Y847	Disease	Likely disease-causing Score 0.545
			**HSF**				**regSNP-intron**		
**rs2297086**	Intronic	-	No significant impact on splicing signals				Likelihood 0.27	FPR 0.56	TPR 0.94
							Benign		

SNV: single nucleotide variant. SIFT: deleterious (0.0–0.05) and tolerated (0.05–1.0); PROVEAN: deleterious ≤ threshold −2.5 and benign > threshold −2.5; Polyphen-2: benign (0.0–0.15), possibly damaging (0.15–1.0), and damaging (0.85–1.0); FATHMM: no significant change (0), unfavorable substitution (<0), and favorable (>0); REVEL: (0–1) higher scores reflect greater likelihood to be disease-causing; MutPred2: * *p*-value 0.0003, ** *p*-value 0.01; HSF: weak splice sites (60–80), average sites (80–90), strong sites (>90); regSNP: for variants’ off-splicing sites: benign (likelihood 0.0–0.56, FPR > 10%, TPR > 52%), possibly damaging (likelihood 0.36–0.45, FPR ≤ 10, TPR ≤ 89%), damaging (likelihood 0.45–1.0, FPR ≤ 5%, TPR ≤ 84%). SIFT: Sorting Intolerant from Tolerant, PROVEAN: Protein Variation Effect Analyzer, Polyphen-2: Polymorphism Phenotyping-2, FATHMM: Functional Analysis through Hidden Markov Models, REVEL: Rare Exome Variant Ensemble Learner. FPR: false positive rate, TPR: true positive rate.

## Data Availability

The original contributions presented in the study are included in the article; further inquiries can be directed to the corresponding author/s.

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
