# Peer review of "PTCH1 Gene Variants, mRNA Expression, and Bioinformatics Insights in Mexican Cutaneous Squamous Cell Carcinoma Patients"

_biology, 2024, doi:10.3390/biology13030191_

Round 1

Reviewer 1 Report

Comments and Suggestions for Authors

The paper is devoted to the elucidation of the PTCH1 gene expression and variability in patients with squamous cell carcinoma. PTCH1 has a know role in the cancer process, so this topic is relevant to the field.
The research expands the field by ading data on this skin cancer type, albeit with a negative result (no association was found). The scientific significance of these results is rather low, as there is little use of knowing that this gene does not participate in the process.
The mehodology, conclusions, and references are appropriate.

1. In the abstract, it would de useful to have some motivation for the studying of this particular gene
2. In the introduction, there is rather little information on the PTCH1 role in the skin cancer, which is the topic of the paper
3. Table 1 format is misleading, as the control group is not separated from the cancer group. I suggest splitting it into two tables.
4. Table 2 is hard to read, as it spans several pags with headers only on the first page. I suggest either to repeat the heading, or to split the table into several smaller ones
5. As far as it goes about the analysis of individual patients, Figure 1 would be easier to analyse in the form of box-and-whiskers plot
6. The discussion does not address the question on why there is such a large difference between BCC and cSCC in respect to the PTCH1 association.

Reviewer 2 Report

Comments and Suggestions for Authors

In this study, the authors compared PTCH1 expression levels and the frequency of 4 SNPs in PTCH1 between blood samples of patients with squamous cell carcinoma of the skin and controls. No differences in distribution of the SNPs or in the expression levels of PTCH1 were observed between groups. As the results were all negative, the authors performed bioinformatic analysis, which are of no value. Major changes and clarifications are needed in order to improve the manuscript.

1. The authors should make clear the rationality of the study. Which is the frequency of PTCH1 mutations or dysregulated expression in squamous cell carcinoma of the skin? The argument that mutations in PTCH1 caused nevoid basal cell carcinoma syndrome is weak.

2. Why were the mRNA levels verified in the peripheral blood leukocytes instead of tumor, which seems the natural site to be explored? What is the connection between PTCH1 mRNA levels in the peripheral blood leukocytes and in the tumor cells that support this approach?

3. The possible roles or predictions for the 4 SNPs should be clearly described in the Introduction in order to understand the biological sense of the association here described. Why were they chosen? Is there any association between them and the regulation of PTCH1 expression?

4. Could the authors explain why they have explored 2 of the SNPs (rs41313327 and rs2297086) after the publication of the article PMID: 38287479? Both can not be considered as genetic variants in the Mexican population, at least, not in this cohort. I do understand that the control group described here is the same previously explored. Make this clear.

5. The authors should provide details about sample size estimation. This is particularly relevant in situations with neg\ative results, because it is not clear whether the association does not exist or the sample size is unpowered. Please provide sample size calculation or determine the power of the sample based on findings. The authors should take in consideration the size of the genetic effects for each SNP, as well as the minor allele frequencies.

6. What were PTCH1 expression levels evaluated only in 44 control subjects and 26 skin patients? How were they chosen? How are they representative of your entire cohort?

7. The authors have described that 25% of the sample was genotyped twice, but the agreement rate is not described. Describe please the genotype call rate too.

8. The statistical approach is very limited. The authors should apply regression analysis taking in consideration age and sex of patients, since groups are statistically different and both are associated with risk of the disease.

9. It is unclear what the authors want to prove with their bioinformatic analysis. The SNPs are well characterized and 2 of them were unvalidated in the available sample. Moreover, rs357564 is described as benign SNV in the ClinVar, whereas rs41313327 is of unclear significance, but it was not associated with hypospadias (PMID: 3427539) and basal cell carcinoma.

10. A paragraph with limitations and strength of the study would help the readers to understand the meaningfulness of the results. Furthermore, the future scope of the present study should be discussed.

Comments on the Quality of English Language

The study has several grammatical mistakes that distract the attention of the reader.

Reviewer 3 Report

Comments and Suggestions for Authors

The authors present the bioinformatics analysis of the RNA level based gene expression in Mexican cancer cell lines and compare it with normal cells for their potential roles in cancer that could lead to future therapeutics. 

the work is original and seems to be robust. Though there are not significant changes in RNA levels, there could be other mechanisms that could play a role in cancer progression. 

the authors can comment on other factors such as mechanics, chemical environments, pH that could be potential alternative explanations. 

Reviewer 4 Report

Comments and Suggestions for Authors

The simple summary and abstract are not informative and contain general and vague results, especially about the bioinformatics methods.

Wouldn't it be better to choose the cSCC and CG groups from the same age and sex ranges?

Please mention the test and results for this: "There were no statistically significant differences regarding the presence or absence of expression of the gene of interest when compared by clinical features such as age, sex, location and size of the lesions, and histopathological classification (data not shown)"

About the bioinformatics tools: this part is not well-defined and well-established. The problem is not clear, it is not clear why these methods were chosen, and no explanation for biophysical characteristics. Just a homology method; the discussion is not clear. These homology methods in addition to biophysical methods for structural and functional predictions are lately well developed but it is used very preliminary here. To me, this part is not meaningful and irrelevant.

Round 2

Reviewer 2 Report

Comments and Suggestions for Authors

I feel that the authors were receptive to my suggestions, but some very important issues were left behind without solving them. Moreover, some of the answers should be incorporated into the revised manuscript, and others should be recognized as a limitation of the study.

Point 2: It is one of the strongest limitations of the study, since there is no connection which supports mRNA quantification in the blood. The current data on PTCH1 mRNA amounts do not allow any clue about its involvement in skin cancers, and should either be excluded or described as a limitation. This is reinforced by the authors’ answer to my question 3. The authors did not consider any logical/putative connection between SNP function and transcript regulation to select the genetic variants. Two unconnected objectives in a study with the aim of exploring PTCH1 in cutaneous SCC.

Point 4: The authors did not state in the revised manuscript that the data of the control group were already published. Citing published data, even their own data, is a good practice of research.

Point 5: Any sample size calculation in studies exploring connection between genetic variants and phenotypes that does not take into consideration the effect size is ignoring the most important effect. The strategy applied focused on the power for detecting the variant allele, and not the difference between groups and the odds for cutaneous SCC occurrence. The sample size calculation should be clearly described in the revised manuscript, which will allow readers to take their own conclusions. Moreover, the authors should describe the source for the frequency of the alleles that was applied in their estimation.

Point 6: The answer provided by the authors should be incorporated into the revised manuscript.

Point 8: The answer is unclear. I am convinced that regression analysis taking in consideration age and sex of patients is the best approach. These two aspects are well-known to influence risk of cutaneous SCC, and differences exist between groups.

Reviewer 4 Report

Comments and Suggestions for Authors

Thanks to the authors for their response. Unfortunately about the second part of the manuscript I am not convinced. Just bringing some references does not mean that the method is relevant to the rest of the paper. Using bioinformatics, this way, does not add much to the study, just makes it more complicated. Homology method mostly propose some candidates for the reasons of the disease and should be validated by experimental or biophysicl modeling methods but the trend of the manuscript is the reverse and you see that no proper conclusion can be made at last. If you conclude that you need more population, so what is the message of this study?

Round 3

Reviewer 2 Report

Comments and Suggestions for Authors

No further comments. The authors have incorporated my suggestions into the revised manuscript.

Reviewer 4 Report

Comments and Suggestions for Authors

Thanks for, somehow, addressing the concern.